# Fitness cost associated with cell phenotypic switching drives population diversification dynamics and controllability

Lucas Henrion [1], Juan Andres Martinez [1], Vincent Vandenbroucke [1], Mathéo Delvenne [1], Samuel Telek[1], Andrew Zicler[1], Alexander Grünberger[2] & Frank Delvigne [1] ✉

Isogenic cell populations can cope with stress conditions by switching to alternative phenotypes. Even if it can lead to increased fitness in a natural context, this feature is typically unwanted for a range of applications (e.g., bioproduction, synthetic biology, and biomedicine) where it tends to make cellular response unpredictable. However, little is known about the diversification profiles that can be adopted by a cell population. Here, we characterize the diversification dynamics for various systems (bacteria and yeast) and for different phenotypes (utilization of alternative carbon sources, general stress response and more complex development patterns). Our results suggest that the diversification dynamics and the fitness cost associated with cell switching are coupled. To quantify the contribution of the switching cost on population dynamics, we design a stochastic model that let us reproduce the dynamics observed experimentally and identify three diversification regimes, i.e., constrained (at low switching cost), dispersed (at medium and high switching cost), and bursty (for very high switching cost). Furthermore, we use a cell-machine interface called Segregostat to demonstrate that different levels of control can be applied to these diversification regimes, enabling applications involving more precise cellular responses.

Cell populations can respond to environmental changes, and to the frequency of these changes, by adjusting their phenotypes through the activation of dedicated gene circuits[1-3]. This phenotypic plasticity holds significant importance in microbial ecology, where the fitness of a cell population depends on a cost–benefit ratio between the sensing machinery needed for the activation and deactivation of a given gene circuit[4] and its activity. Therefore, controlling the phenotype of cells has a lot of importance in various fields of research, such as bioproduction and synthetic biology, where coordinated gene expression is typically desired[5-10]. Generating and controlling cell collective behavior is considered as a hallmark of synthetic biology[9,11,12], and is now

enabled by the parallel advances made at the level of cell cultivation procedures (i.e., microfluidics[13] and cell–machine interfaces[14]), as well as the manipulation of synthetic gene circuits[15-17]. Effective control of gene expressions and their underlying cellular functions can be achieved in cell populations[5,18,19] or individual cells within a population[20,21]. Different approaches can be used to coordinate/synchronize gene expression in cell populations. On the one hand, specific gene circuits can be designed in order to generate natural oscillations[12,22]. On the other hand, external forcing can be used for coordinating cellular responses[6,23,24]. According to this last approach, a given stimulus (e.g., chemical inducer[20] and light[18,25]) is repeatedly

[1]Terra Research and Teaching Centre, Microbial Processes and Interactions (MiPI), Gembloux Agro-Bio Tech, University of Liège, Gembloux, Belgium.
[2]Microsystems in Bioprocess Engineering, Institute of Process Engineering in Life Sciences, Karlsruhe Institute of Technology, Karlsruhe, Germany.
✉e-mail: F.Delvigne@uliege.be

applied at a given frequency and amplitude in order to entrain gene expression within a cell population. In this case, the effective transfer of information from the extracellular environment to the effector sites within cellular systems is of critical importance and can be corrupted by biological noise[26–28]. In silico experiments pointed out that specific environmental fluctuation frequencies could significantly reduce stochasticity in cell switching[1,28]. This switching behavior, which we will now refer to as a diversification regime, significantly impacts the population structure and lies at the core of phenotypic control. However, the main factors affecting these diversification regimes are not known.

In this work, we experimentally investigate this feature by looking at the temporal diversification profile of different types of cell populations in continuous culture. For this purpose, chemostat runs are complemented by experiments conducted in Segregostat. Segregostat relies on a cell-machine interface to generate environmental perturbations that are compatible with the diversification rate of the considered cell population (Fig. 1)[19]. This rational environmental forcing allows for the observation of several diversification cycles in one experimental run (Fig. 1). We apply this technology to look at the dynamics of cell populations with cellular functions leading to different fitness costs, i.e., utilization of alternative carbon sources (*Escherichia coli*), general stress

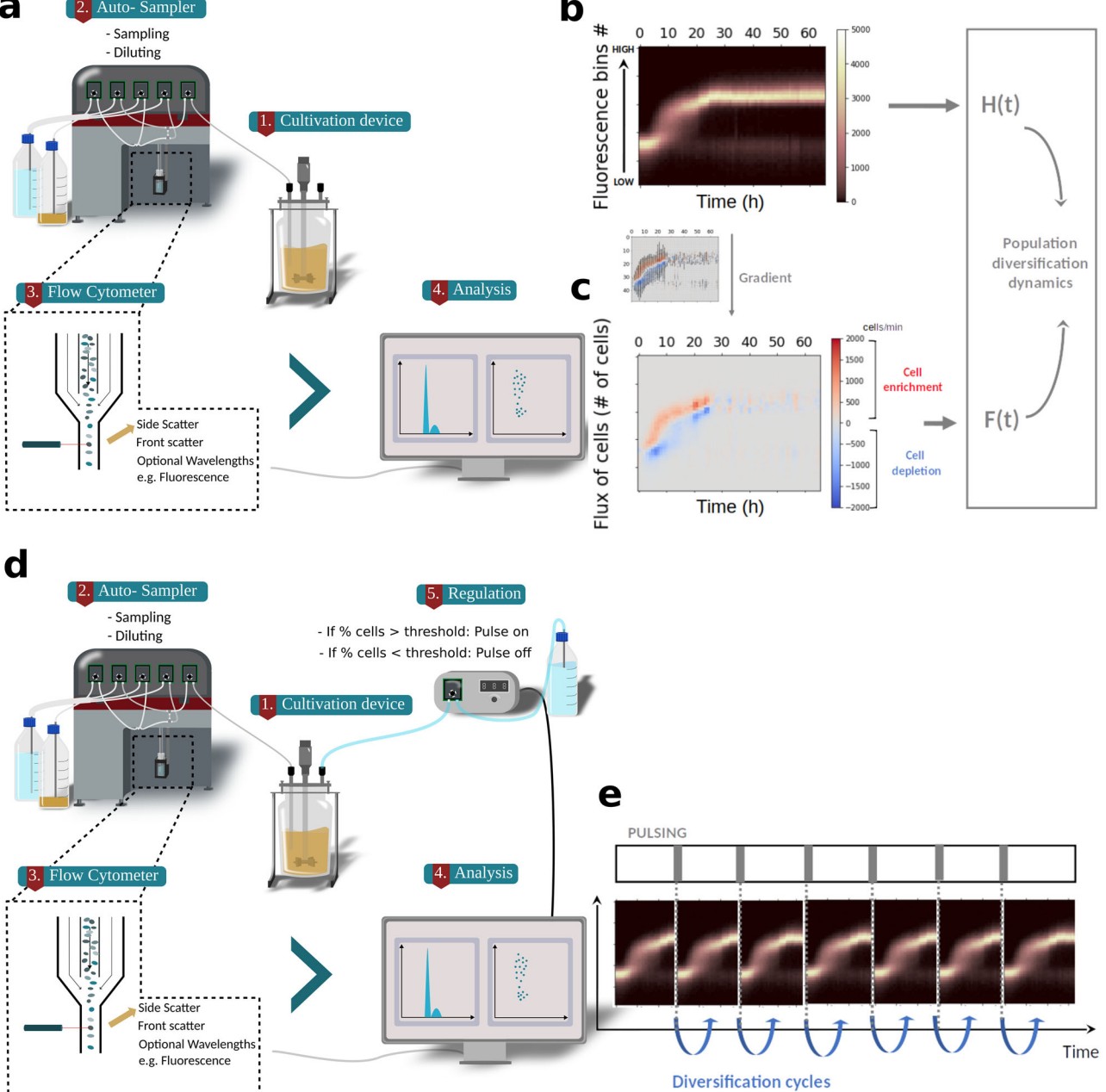

**Fig. 1 | Characterizing cell population diversification dynamics based on automated FC. a** Chemostat culture is monitored based on automated FC. **b** The fluorescence distribution acquired by FC is assembled into a time scatter plot. This time scatter plot is then further reordered into 50 fluorescence bins in order to compute the evolution of the entropy H of the population (Supplementary Note 1). **c** The binned data are further processed by applying a gradient to compute the fluxes of cells into the phenotypic space, leading to the quantification of the total fluxes of cells per time interval *F(t)*. Both *H(t)* and *F(t)* will be exploited for characterizing the phenotypic diversification dynamics of diverse cell populations. **d** Scheme of Segregostat set-up. Pulses of nutrients are added in function of the ratio between GFP negative and GFP positive cells, as recorded by automated FC. **e** Expected evolution of a Segregostat experiment where, upon controlled environmental forcing, several diversification cycles can be generated.

response (*E. coli* and *Saccharomyces cerevisiae*), sporulation (*Bacillus subtilis*) and activation of a T7-based expression system (*E. coli*). Based on the fitness cost associated with the cell switching mechanism (referred to as switching cost or fitness cost in this study), three different population diversification regimes, with different levels of sensitivity to environmental perturbations, are observed.

## Results

### Characterization of population diversification dynamics based on automated flow cytometry

In the context of this work, the temporal diversification of cell populations has been followed based on chemostat cultivation of GFP reporter-bearing strains and automated flow cytometry (FC) (Fig. 1a). We define the phenotype as the cellular content in GFP and the characteristics associated with the activation of the observed gene circuit. By coupling FC to GFP, we are able to visualize the diverse range of phenotypes and study the dynamics of population diversification based on snapshot data. To better describe these dynamics, we have developed a methodology to compute the fluxes of cells from one phenotype to another and the resulting degree of heterogeneity of the population, i.e., in our case, based on the measurement of information entropy (Fig. 1b) and the flux of cells (Fig. 1c). Entropy is a measurement derived from information theory allowing to compute the degree of heterogeneity of a population[29]. Briefly, GFP distributions obtained from automated FC measurements are binned and the resulting phenotype distribution is used to compute the population entropy (Supplementary Note 1 and Supplementary Fig. 1). Based on the same binning strategy and by evaluating the enrichment or depletion of bins between consecutive measurements, we can also determine the flux of cells. Consequently, we can assess the dynamics of population diversification by monitoring the changes in population diversity ($H(t)$), as well as understanding how the phenotype shift evolves over time ($F(t)$). In addition to quantifying cell-to-cell heterogeneity within the population, $H$ will also be used to calculate the information transmission to a cell population when subjected to environmental forcing. The benefit of this proxy is its independence from the mean of the distribution, by contrast with other noise proxies (e.g., Fano factor) that are known to be overestimated when the mean value increases[30,31]. Using entropy to analyze chemostat experiments, however, provides limited information about diversification dynamics. Indeed, the main diversification process takes place during the transition between the batch and continuous phases of the culture. Therefore, we used a cell–machine interface allowing us to produce several diversification cycles in a single experiment. This device is called Segregostat and comprises a continuous cultivation device connected to an in-house online FC platform[19] (Fig. 1d). This device enables the generation of several diversification cycles per experiment, leading to a better characterization of the population switching dynamics (Fig. 1e). Practically, the cells analyzed based on automated FC are clustered into a GFP negative and a GFP positive group. Depending on the gene circuits used, a pulse of inducer is applied when the minimum ratio between the two phenotype clusters (i.e., either 50% or 20% of the total amount of cells in the desired state, depending on the cellular system considered) is not reached. Based on this experimental and theoretical framework, we characterized the population diversification profile of six different gene circuits in three distinct cellular systems. Our approach involved linking a GFP reporter to each gene circuit, enabling us to leverage the analytical power of FC (20,000 cells per analysis) to study the population diversification over time.

### Coordinated gene expression in a cell population can be obtained based on environmental fluctuations triggered by a cell–machine interface

The methodology described in the previous section was applied to map the diversification of cell populations in chemostat and Segregostat cultivations. We began our analysis by considering two gene circuits involved in simple cellular processes in *E. coli*, i.e., the activation of the arabinose (Fig. 2a) and lactose (Fig. 2b) operons, respectively. Extended analysis of these systems is provided in Supplementary Fig. 2, and reproducibility of the data is displayed in Supplementary Fig. 3. This type of cellular process is quite simple since it involves two inputs, i.e., the absence of glucose or its limitation, and the presence of either lactose or arabinose as an alternative carbon source[4]. Since these cultures were conducted in continuous mode, it was quite easy to ensure glucose limitation. Furthermore, the gene circuitry behind the activation/deactivation of these two operons is well documented in the literature[4,32], making them ideal case studies.

The maximal growth rate of these alternative carbon sources is close to the one of glucose. Further, despite the production of GFP, the induced phenotype still has a maximal growth rate above the dilution rate imposed in continuous cultivation. Thus, a feature of these two systems is that their activation does not result in a reduction in growth rate in these conditions, which we will from now on refer to as a fitness reduction or switching cost. An extended analysis of the switching cost for the different systems can be found in Supplementary Note 2 and Supplementary Figs. 4 and 5. We thus decided to investigate other systems involved in more complex cellular processes known to lead to a higher switching cost. We first chose to consider the general stress response in *E. coli* and selected the promoter of the *bolA* gene as a representative $\sigma^S$-dependent system[33,34] (Fig. 2c). To extend our analysis to another biological system, we also selected a gene circuit involved in the accumulation of glycogen in yeast[35], i.e., P*glc3*, as a representative reporter of bet-hedging in *S. cerevisiae*[36,37] (Fig. 2d). Both genes are involved in very complex regulons, making it difficult to find an external trigger. However, these general stress response reporter systems are known to share common features in the sense that their expression is anticorrelated with the growth of individual cells, making them very useful for analyzing cell collective behavior such as bet-hedging[34,38]. We then decided to use the external glucose concentration as the main actuator for these two systems. Glucose-limited chemostats were then run as reference conditions. For Segregostat experiments, glucose was pulsed instead of lactose or arabinose, allowing it to generate feast-to-famine environmental transitions.

Segregostat cultivation of all four cellular systems investigated led to entrainment and sustained oscillation of gene expression (Fig. 2a–d). Based on the analysis of the entropy $H(t)$ and the flux of cells $F(t)$ over time, we observed that for all systems, entrainment phases were accompanied by an increased flux of cells switching to the alternative phenotype and a corresponding decrease of entropy $H(t)$ at the time of pulsing (Fig. 2e where the analysis is shown for the P*araB*::GFP system, Supplementary Note 1, Supplementary Fig. 2). However, the entropy increases during the relaxation phase (GFP dilution upon cell division).

In addition to these four gene circuits, two more were analyzed: a T7-based expression system in *E. coli* and a circuit involved in sporulation in *B. subtilis*. These two systems exhibit a very high switching cost, leading to very specific population diversification profiles that will be described in the next section.

### Phenotypic switching associated with extreme fitness cost gives rise to a bursty diversification regime

We investigated the diversification dynamics of two other systems known for their high impact on cellular fitness; the T7-based expression system in *E. coli* BL21 (pET28:GFP)—a typical heterologous protein production platform—and the sporulation regulon in *B. subtilis* (P*spoIIE*:GFP). The T7 expression system is inducible by lactose and P*spoIIE*:GFP is expressed when glucose is limiting, thus acting as an early trigger for sporulation. The pET28:GFP system is a T7-based expression vector inducible with lactose. When lactose is pulsed, cells are

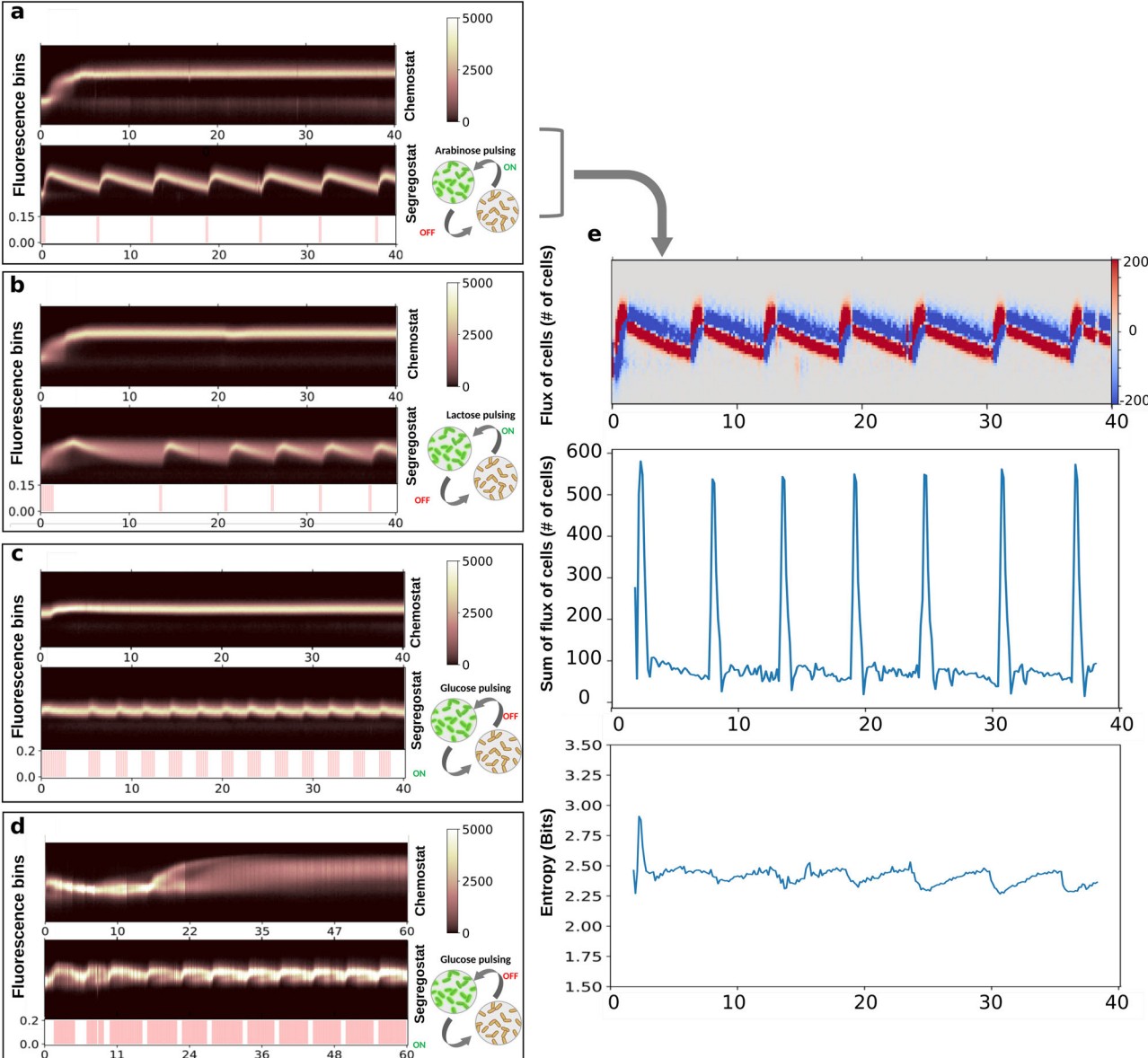

**Fig. 2 | Population diversification dynamics recorded based on automated FC for four different cellular systems.** Time scatter plots (in hours) binned into 50 cell clusters (fluorescence bins) for cultivations made in chemostat and Segregostat for **a** the P$_{araB}$::GFP system in *E. coli*. **b** the P$_{lacZ}$::GFP system in *E. coli*. **c** the P$_{bolA}$::GFP system in *E. coli*. **d** the P$_{glc3}$::GFP system in *S. cerevisiae*. Animated movies for the time evolution of the FC raw data for each system are available. For Segregostat experiments, environmental forcing has been performed based on nutrient pulsing (the type of nutrient shown in the drawings for cell switching). **e** Computation of

the flux of cells and the entropy (higher values mean more heterogeneous) for the P$_{araB}$::GFP system cultivated in Segregostat mode (Supplementary Note 1, Supplementary Fig. 2). Reproducibility of the data has been assessed based on two biological replicates (*n* = 2) for each system. All FC measurements contain 20,000 analyzed cells (Supplementary Note 1 and Supplementary Fig. 3). The color bar alongside the scatterplot represents the number of cells. Source data are provided as a Source Data file.

turning green, and GFP is diluted by growth when lactose pulsing is turned off (glucose is added continuously according to a classical continuous mode of cultivation at a dilution rate $D = 0.5\,h^{-1}$). The P$_{spoIIE}$:GFP system is induced upon glucose limitation. In our case, this system is cultivated at a very low dilution rate ($0.1\,h^{-1}$) in order to generate glucose limitation and stimulate cell switching to sporulation. When too many cells are switching, additional glucose is pulsed in order to keep the population under control. For both, the phenotypic switch leads to a drastic growth reduction. Surprisingly, even in chemostat (with lactose present for the T7 system), FC profiles reveal bursts of diversification (Fig. 3a, b) and a subsequent high entropy at the population level. These bursts are the result of a subpopulation of cells deciding to switch and being washed out from the continuous

cultivation device due to the associated fitness cost. Then, upon environmental forcing based on Segregostat cultivation (lactose pulse, if more than 50% of cells are not induced with the T7 system and glucose, is pulsed if more than 20% of cells express P$_{spoIIE}$:GFP), the number of bursts is reduced, and the fluxes of cells involved in the process are increased, leading to a substantial but temporary reduction of the entropy for the population (Fig. 3a, b). These results point out that Segregostat transiently reduces the average entropy of gene circuits with a high fitness cost despite very complex dynamics. It has been suggested in the literature that the stochasticity in cell switching is associated with its associated fitness cost and is important for the survival of the whole population. This feature is well illustrated in this case, where a phenotype switch induces a dramatic loss of growth rate,

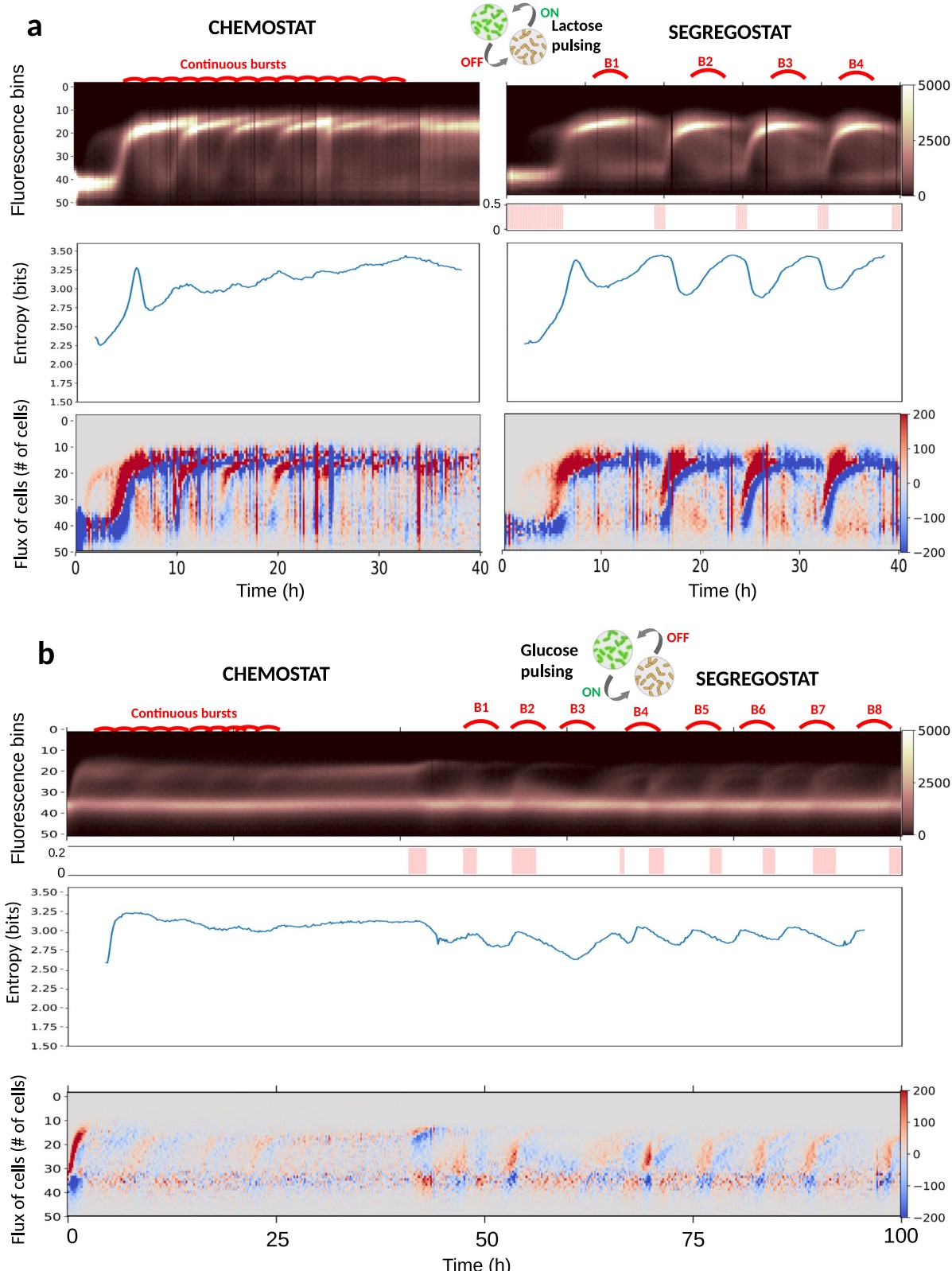

**Fig. 3 | Population dynamics for two cellular systems exhibiting bursts in diversification.** Temporal diversification profile for the $P_{T7}$::GFP system in *E. coli* **a** and the $P_{spoIIE}$::GFP system **b** cultivated in chemostat (from 0 to 40 h for $P_{spoIIE}$::GFP) and Segregostat modes (from 40 to 100 h for $P_{spoIIE}$::GFP). Below Segregostat cultivation modes, vertical lines indicate times of pulsing with the concentration (in g/L) displayed on the *y*-axis. Only the continuous phase of the cultivation is shown on the graphs. Bursts of diversification are highlighted in red on the fluorescence bins data. In both cases, the entropy $H(t)$ and the fluxes of cells in the phenotypic space $F(t)$ have been computed from the binned fluorescence data. The reproducibility of the data has been assessed based on two biological replicates ($n = 2$) for each system. For each time interval, 20,000 cells have been analyzed by FC (Supplementary Note 1 and Supplementary Fig. 3). The color bars represent the number of cells for the fluorescence time scatter plots and the number of cells per minute for the $F(t)$ scatter plots. Source data are provided as a Source Data file.

leading to the wash-out of these cells in continuous cultivation conditions. However, this stochasticity can be reduced by applying environmental perturbations at a rate matching the phenotypic switching rate of cells. In the context of the T7 system, this approach led to the periodic maximization of cells in the GFP-positive state, suggesting that it could be used for mitigating metabolic burden and maximizing productivity in continuous bioprocesses. Indeed, the flux of cells to the high fluorescence bins is more regular for Segregostat conditions (Fig. 3a).

## Coordinated gene expression does not necessarily lead to a more homogeneous cell population

Based on the analysis of the population diversification profiles exhibited by the six systems investigated, both similarities and differences can be observed. All six systems exhibit coordinated gene expression upon environmental forcing in the Segregostat device. This feature can be notably quantified based on $F(t)$. On the other hand, the same systems display different $H(t)$ profiles (Fig. 4). We computed the average entropy $H(t)$ for the six systems upon cultivation in chemostat and observed systems exhibiting a low basal entropy of around 2.2 bits (i.e., $P_{araB}$::GFP, $P_{lacZ}$::GFP and $P_{bolA}$::GFP) and systems exhibiting a higher basal entropy of more than 3 bits (i.e., $P_{glc3}$::GFP, $P_{T7}$::GFP and $P_{spoIIE}$::GFP) (Fig. 4a). For each system, the impact of Segregostat control system on the entropy was then investigated. We use entropy as an indicator of control because one would expect a controlled system to exhibit a more homogeneous phenotype and, therefore, a lower $H(t)$. However, based on this definition, we observed that not all systems were controllable with Segregostat, as some of them showed no decrease in entropy despite their coordinated gene expression. In particular, (1) the systems exhibiting a low basal entropy ($P_{araB}$::GFP, $P_{lacZ}$::GFP, and $P_{bolA}$::GFP) showed no decrease in entropy. $P_{lacZ}$::GFP even showed an increased entropy instead, probably due to the leakiness of the promoter during the relaxation phase of the diversification cycles (Fig. 4b, c). 2) The systems exhibiting a high basal entropy ($P_{glc3}$::GFP, $P_{T7}$::GFP and $P_{spoIIE}$::GFP) showed a homogenization of the population. (Fig. 4b, c).

We thus propose this criterion as a classification of the cellular systems, where trying to control a homogeneous cell population in chemostats produces no benefits and can even lead to an increased heterogeneity, while heterogeneous systems in chemostats can, by contrast, be made much more homogeneous (Fig. 4c).

Based on this criterion, it can be observed that the $P_{glc3}$::GFP system in *S. cerevisiae* exhibits a higher level of controllability than the other systems. Indeed, in this case, Segregostat cultivation led to a drastic decrease in $H(t)$ by comparison with the reference condition in chemostat. This effect was then verified in microfluidics. Stress response pathways in yeast are known to be involved in bet-hedging strategies, leading to a trade-off between growth and expression of stress-related genes[37,38]. The *glc3* gene belongs to this category. Accordingly, cells activating *glc3* exhibit reduced growth. This phenomenon has been characterized based on microfluidics single-cell cultivation (MSCC)[39] experiments allowing to exposure of yeast cells to tightly defined glucose concentrations (Supplementary Fig. 4). Unlike with FC analyses where only population snapshots are captured, these experiments let us monitor cell traces and thus analyze the fitness cost (growth reduction upon switching) associated to the switching. At low glucose concentration (<0.2 mM), a single cell fully activates the stress reporter and stops growing (Supplementary Movie 1). At a higher glucose concentration, the growth of the microcolonies is faster, but some stochastic switching events can be clearly observed, with cells suddenly expressing the fluorescence reporter and stopping their growth (Supplementary Movie 2). In order to confirm the beneficial impact of Segregostat condition on the $P_{glc3}$::GFP, we used dynamic microfluidic single-cell cultivation (dMSCC)[40] where we applied environmental fluctuations between 0.1 and 1 mM of glucose at the

frequency recorded in Segregostat conditions. These fast and sharp transitions are an idealized scenario of Segregostat cultivation, but similarly, we observed a very homogenous gene expression pattern with cells turning green in perfect synchrony (Fig. 4d, Supplementary Movie 3). The growth of all cells was comparable; the stress level was kept at a low-level thanks to the fluctuating environmental conditions. If we compare the experiments run in MSCC and dMSCC, it can be observed that cultivated cells under fluctuating glucose concentration lead to an intermediate scenario, both at the level of the mean fluorescence profile (Fig. 4e) and the global entropy (Fig. 4f). It seems that when phenotypic switching is associated with a loss of fitness, there is more stochasticity in the diversification pattern followed by the population. However, when the nutrient level is changed at a given frequency, switching and growth can be kept under control, leading to a drastic reduction of the phenotypic heterogeneity of the cell population. Another parameter that can explain the differences observed between the $P_{glc3}$::GFP system and the $P_{T7}$::GFP and $P_{spoIIE}$::GFP systems (Fig. 4b, c) is the rate of switching. It can be reasonably assumed that gene expression and cell growth are slower in Eukaryotes than in Procaryotes[41], potentially explaining why it is easier to maintain the $H(t)$ profile at a very low level in Segregostat conditions for the $P_{glc3}$::GFP system since cells are switching more slowly in this case.

The concept of controllability refers to the extent to which environmental forcing provided based on Segregostat cultivation reduces or not the global entropy of the population. So far, we have used information entropy as a proxy for the quantification of the heterogeneity of cell populations. However, information entropy can also be used to evaluate the mutual information (MI), i.e., the reduction in entropy of the cell population when appropriate external stimulations are applied (Supplementary Note 3, Supplementary Figs. 6–9). We then computed MI for a system exhibiting low ($P_{araB}$:GFP) and high ($P_{glc3}$::GFP) controllability (Supplementary Note 3, Supplementary Fig. 10). Based on this analysis, we concluded that the MI is already maximal in chemostat conditions for the $P_{araB}$::GFP system, explaining the low controllability measured in this case.

## Fitness cost drives the appearance of different dynamical regimes with different levels of controllability

According to the datasets acquired from the different biological systems, we observed three types of diversification regimes named respectively constrained, dispersed and bursty (Fig. 5). We proceeded to an in-depth analysis of the potential factors influencing the three observed modes of diversification. The key distinguishing factor between the systems displaying low and high basal entropy and controllability lies in the fitness cost linked to phenotypic switching. When the fitness cost associated with the switching is low or non-existent, we observe a constrained diversification regime where the population switches upon environmental change and adopts a homogeneous distribution. In that case, the stimulation of the population with controlled environmental pulsing does not homogenize it (since information transmission is already maximal in the non-controlled conditions). When there is a fitness cost associated with the switch, what we call a dispersed diversification regime can be observed. In that case, cells react to environmental changes but then adopt a more heterogeneous population structure. In this case, the application of controlled environmental perturbations allowed a substantial reduction in population heterogeneity. For the bursty diversification regime (higher fitness cost), cells switch in bursts, leading to a very heterogeneous population structure. The application of controlled environmental perturbations reduces the number of bursts and increases the number of cells involved in these bursts, leading to a transient decrease in population heterogeneity.

It seems that the regime depends on the fitness cost associated with the phenotypic switching event. In order to verify that fitness cost is indeed the driver for the diversification pattern adopted by cell

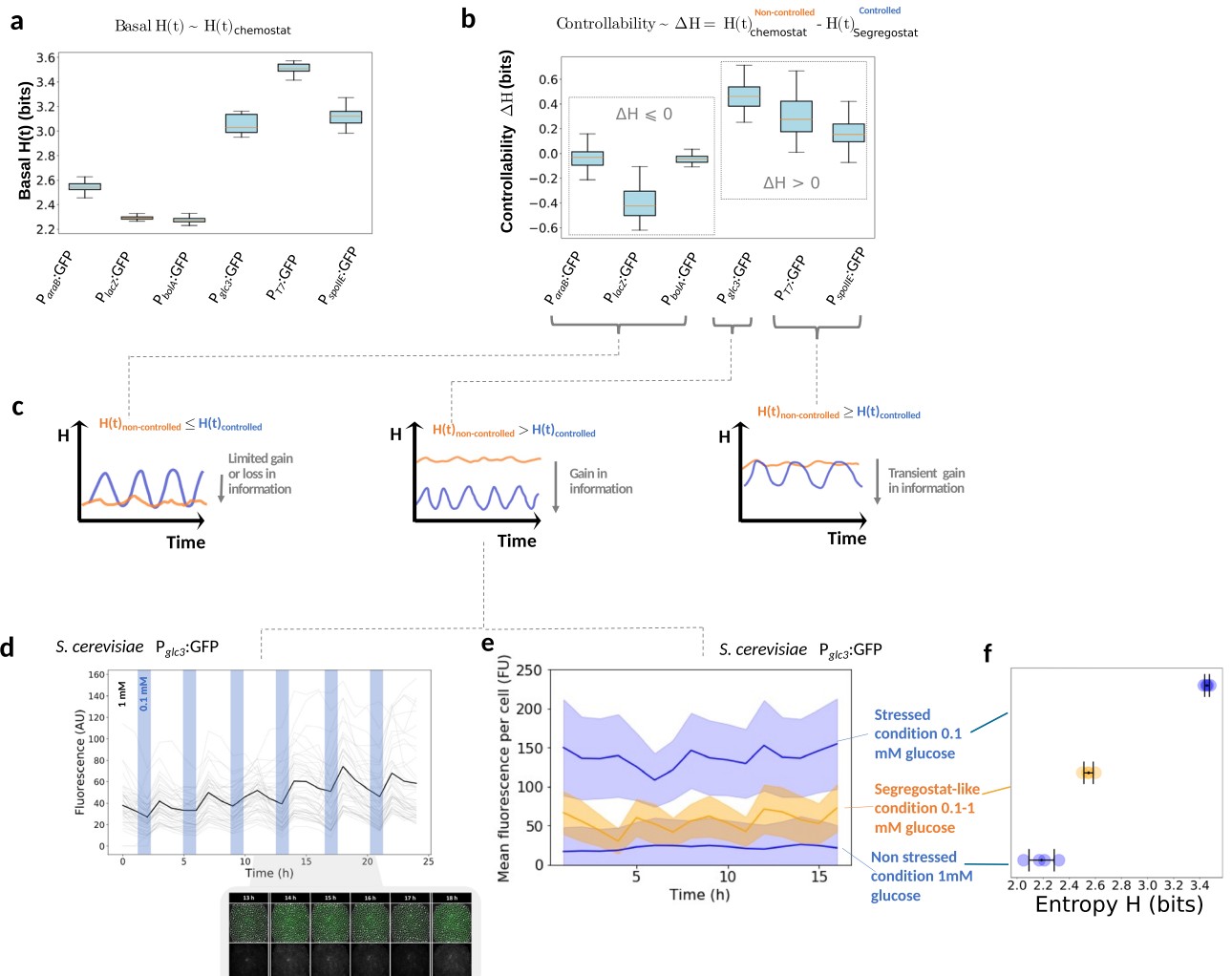

**Fig. 4 | Computation of controllability based on the entropies for the different cell systems and cultivation devices used. a** Basal entropy (represented as a boxplot with whiskers where the interquartile range as extremities of the box, the median is the horizontal line in the box, and the whiskers as data extremes) recorded based on automated FC during chemostat experiments for the six biological systems investigated (values of entropy are computed from the population diversification profiles over the entire cultivation). Experiments have been done in duplicates ($n$ biological replicate = 2) for each cellular system and cultivation conditions (chemostat or Segregostat) and exhibit a high reproducibility, see Supplementary Fig. 3). **b** Computation of the controllability (gain in entropy from chemostat to Segregated conditions represented as a boxplot with same structure than for plot A) for the six biological systems investigated. These average gains have been obtained by subtracting the mean value of $H(t)$ recorded in Segregostat (considered as the controlled condition, i.e., cell population under environmental forcing) from the basal entropy. **c** Different diversification profiles, with different levels of controllability, can be observed based on the comparison of the H(t) profiles between non-controlled (chemostat) and controlled (Segregostat)

conditions. **d** Single-cell traces of yeast P$_{glc3}$::GFP cells cultivated in a dMSCC device fluctuating between 1 and 0.1 mM of glucose ($T_{1mM}$ = 3 h; $T_{0.1mM}$ = 0.8 h). Between 10 and 40 cells have been tracked in four different cultivation chambers over two biological replicates (mean fluorescence is shown in bold). Pictures of a microcolony taken at regular time intervals are shown (Supplementary Movie 3). **e** Comparison of the mean fluorescence profile obtained in dMSCC with the ones obtained in classical MSCC at high (1 mM) and low (0.1 mM) glucose concentrations. The shaded region around the lines represents the standard deviation computed from the measurement (between 10 and 40 cells have been tracked in four different cultivation chambers over 4 biological replicates ($n$ = 4) for each condition. **f** Mean values of the entropy over the whole microfluidics experiments run at different glucose concentrations and standard deviation across chambers ($n$ microfluidic chambers = 4 observed over one experiment). Each mean value over a cultivation run in a chamber is represented as a dot, and the mean of all replicates with the standard deviation across them as a black dot with error bars. Source data are provided as a Source Data file.

populations, we conducted in silico experiments. For this purpose, we considered the kinetic parameters obtained from the inference of the P$_{glc3}$::GFP system in yeast and conducted stochastic simulations based on FlowStocKS (Supplementary Note 4). We conducted 32 different chemostat simulations by varying only the value for the fitness cost and computed the entropy $H$ (Fig. 6a) and the fluxes of cells ($F$) involved in phenotypic switching (Fig. 6b). Solely based on the fitness cost associated with the switching, we were able to reproduce the three types of diversification regimes experimentally observed during the experiments (Fig. 6c). Complete wash-out of the cells was observed

for extreme fitness cost (>99% reduction in growth rate). We then wondered if we could observe clear transitions between the different regimes. Such transition was observed between the bursty and the dispersed regime based on the computation of the flux of cells $F$. Indeed, while the transition between the dispersed and constrained regime is progressive (Fig. 6d), the bursty regime is marked by the appearance of a strong variation in flux of cell which is not observed for the other two regimes (Fig. 6e). FlowStocKS was also able to reproduce the behavior of the population under Segregostat cultivation, and the reduction in entropy upon environmental forcing was

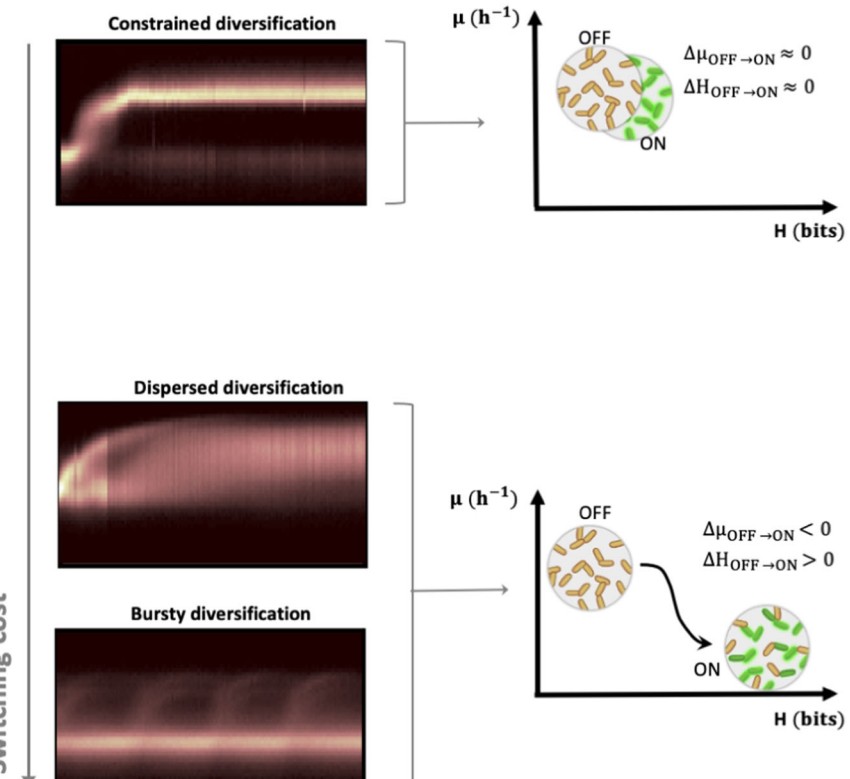

**Fig. 5 | Illustration of the three diversification regimes observed based on automated FC in the function of the switching cost.** The constrained diversification regime is observed at low switching costs. According to the regime, all cells switch from the OFF state to the ON state according to a relatively homogenous diversification process, and the population exhibits a low $H(t)$ (there is no increase in entropy upon activation of cells. The dispersed and bursty regimes are observed at high switching costs. Accordingly, cells switch from the OFF to the ON state in a quite heterogeneous way. The resulting stochastic switching is hypothesized to play a role in the stabilization of the cell population under continuous cultivation as a way to mitigate the fitness cost associated with switching. Upon diversification, the population exhibits a higher entropy.

computed (Fig. 6f). Again, reduction in entropy depended on the associated fitness cost and thus was observed for the dispersed and bursty regimes, in accordance with our experimental observations.

## Discussion

We used Segregostat to better characterize cell population diversification dynamics by generating successive diversification cycles during the same experimental run. The basic principle behind this technology is to revert the environmental conditions when a fraction of cells (50% or 10% of the total population, depending on the investigated system) crosses a predefined fluorescence threshold. This approach allows to maintain a cell population in a dynamic switching state during the experiment. Based on the analysis of the MI, i.e., the amount of information transferred from the extracellular conditions to the cell systems[27,29,42], we determined that for the $P_{araB}$:GFP system, the chemostat drives a similar amount of information to the Segregostat. On the other hand, we observed a drastic reduction in entropy when entraining stress-related systems, such as the $P_{glc3}$:GFP system in yeast, in Segregostat. In this case, we determined that the high entropy observed in the chemostat was related to a trade-off between growth and gene expression[36–38,43–45], which was further confirmed based on a microfluidics experiment. To better relate the switching cost to the resulting population structure, we then considered two additional systems where phenotypic switching induced a huge fitness cost i.e., the sporulation system in *B. subtilis* and the T7-based expression system in *E. coli*.

Based on all the data accumulated by automated FC for six different biological systems, we found that cell populations diversified according to three distinct regimes associated with increasing fitness costs, i.e., constrained, dispersed, and bursty. The most noticeable difference between these regimes is the level of entropy of the cell population, the entropy being a measure derived from information theory giving a robust estimate of population dispersion[29,46]. The lowest entropy values were associated with the constrained regime and the highest ones were seen for dispersed and bursty regimes. The other difference was observed in the cultivation of cell populations under fluctuating environmental conditions. In Segregostat, a reduction of entropy compared to chemostat cultivation was associated with the dispersed and bursty regimes but not the constrained one. Taken altogether, the data suggested that on top of affecting the population heterogeneity, the phenotypic switching mechanism changes the controllability of the system. All these observations were confirmed based on stochastic simulations (FlowStocKS), suggesting that the proposed diversification framework could be generalized for characterizing diversification dynamics for any kind of cellular system.

Harnessing phenotypic heterogeneity of microbial populations has been the subject of much research, leading to the design of various technologies aiming at homogenizing gene expression in cell populations[47]. We have shown that the level of diversification of microbial populations cultivated in continuous bioreactors depends mainly on the fitness cost. Since many applications involve gene circuits whose activation leads to a substantial burden for the cellular system[48,49], active diversification processes have to be expected in a number of cases[50]. For example, bursty diversification profiles have been observed for two cellular systems exhibiting high switching costs. According to this regime, marked cycles of diversification can be observed even in chemostat cultures. These cycles are due to the rapid switching (burst) of a fraction of the population that lower the average

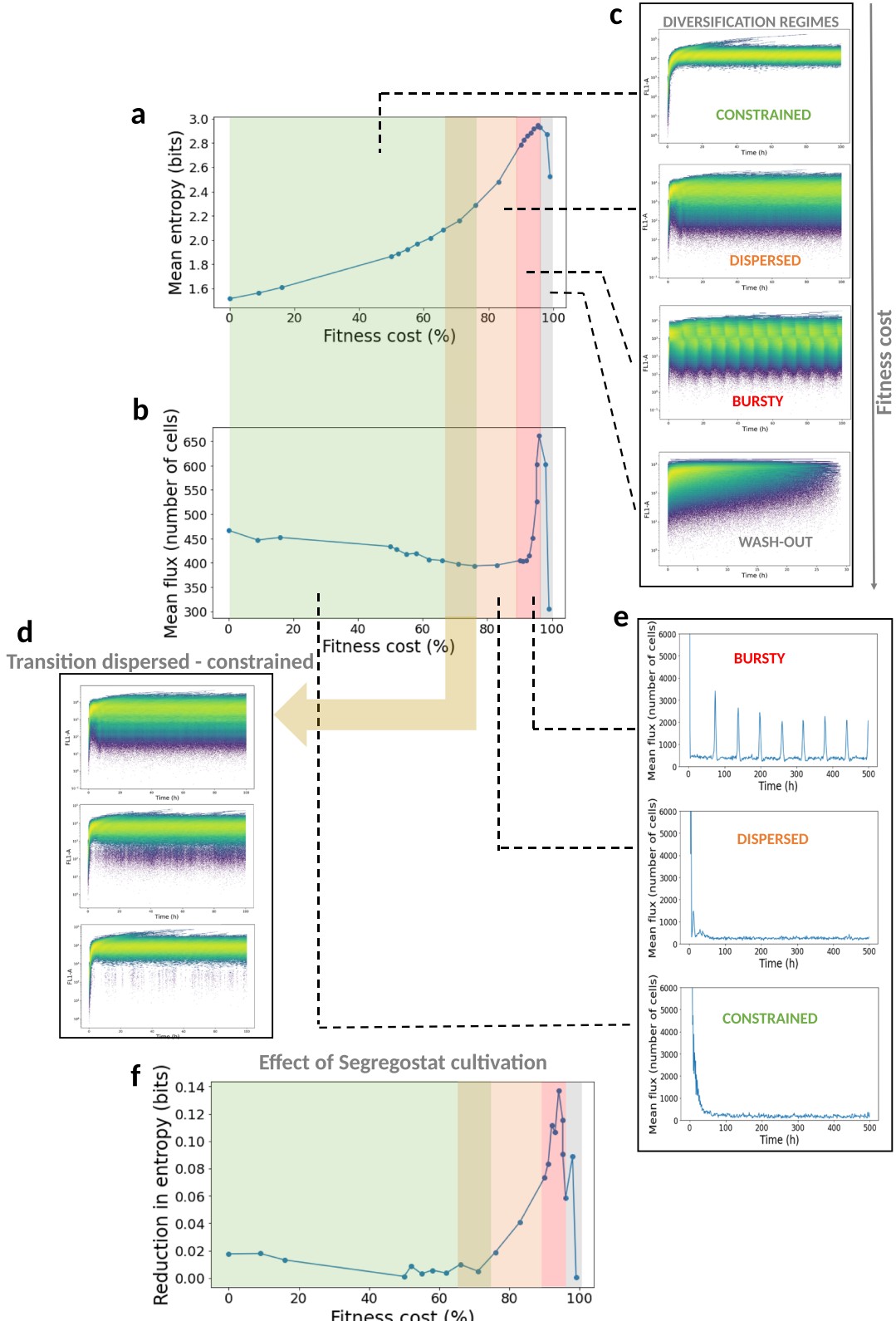

growth rate of the population and are washed out of the system. These cells are then replaced by the next burst of diversification, starting from a subpopulation of non-diversified cells. This type of temporal profile has been previously observed, but it is based on spatially organized cells equipped with synthetic circuits[7,51,52]. Here, we show that it is possible to reproduce such a complex but organized diversification profile with cells in suspension in a bioreactor and that the

complex dynamics behind phenotypic heterogeneity are linked to the fitness cost associated to the switch.

## Methods

### Stains and plasmids
The analyses of alternative carbon source utilization and stress response in *E. coli* were done based on a *E. coli* W3110 backbone and

**Fig. 6 | Main outputs of the FlowStocKS simulations. a** Evolution of the mean entropy recorded from the H(t) profile in a chemostat in function of the fitness cost (here expression as the percentage of reduction of the initial growth rate prior phenotypic switching). These values are equivalent to the basal entropy shown in Fig. 4. **b** Evolution of the mean flux of cells recorded during chemostat experiments in the function of the fitness cost associated with phenotypic switching (see Fig. 1 for more details about the computation of the flux of cells). **c** Selected simulated time scatter fluorescence plots illustrating the different diversification regimes observed at different fitness costs (the whole simulation dataset can be found in Supplementary Movie 4). **d** Selected simulated time scatter fluorescence plots

illustrating the progressive transition between the dispersed and constrained diversification regimes. **e** Selected simulated time evolution for the fluxes of cells recorded for different values of fitness cost. The bursty regime is characterized by the spontaneous generation of flux of cells (bursts) in chemostat cultivations. **f** Reduction of the entropy upon environmental forcing in the function of the fitness cost associated with phenotypic switching. The entropy values have been computed by subtracting the mean entropy value recorded for the chemostat experiment from the corresponding ones obtained in Segregostat and are equivalent to the controllability shown in Fig. 4.

kanamycin resistance bearing plasmids originating from the Zaslaver collection (i.e., $P_{araB}$::GFPmut2, $P_{lacZ}$::GFPmut2 and $P_{bolA}$::GFPmut2)[53]. For investigating the T7 induction system response, we used *E. coli* BL21 (DE3) carrying pET28:GFP[54]. To observe the starvation response, we used *S. cerevisiae* CEN.PK 113-7D background with the chromosomal integration of a reporter cassette $P_{glc3}$:eGFP[35,55]. The *E. coli* W3110 *ΔaraBAD* strain used for determining the conditional entropy of the $P_{araB}$:GFP system was constructed through CRISPR–cas9 enhanced lambda red phage mediated homologous recombination[56]. The primer sequences are described in Supplementary Table 1. Finally, we monitored the early stage of the sporulation process in *B. subtilis* 168 based on a chromosomal integration of $P_{spoIIE}$::GFPmut2 (kindly provided by Denise Wolf and Adam Arkin)[57].

## Cultivation conditions and segregostat procedure

Bacteria (*E. coli* and *B. subtilis*) precultures and cultures have been performed in a defined mineral salt medium containing (in g/L): $K_2HPO_4$ 14.6; $NaH_2PO_4·2H_2O$ 3.6; $Na_2SO_4$ 2; $(NH_4)_2SO_4$ 2.47; $NH_4Cl$ 0.5; $(NH_4)2$-H-citrate 1; glucose 5, thiamine 0.01, antibiotic 0.1. Thiamine is sterilized by filtration (0.2 mg/L). The medium is supplemented with 3 mL/L of a trace element solution, 3 mL/L of a $FeCl_3·6H_2O$ (16.7 g/L), 3 mL/L of an EDTA (20.1 g/L) and 2 mL/L of a $MgSO_4$ solution (120 g/L). The trace element solution contains (in g/L): $CoCl_2·H_2O$ 0.74; $ZnSO_4·7H_2O$ 0.18; $MnSO_4·H_2O$ 0.1; $CuSO_4·5H_2O$; $CoSO_4·7H_2O$. Filtered sterilized kanamycin (50 mg/L) was added for plasmid maintenance in *E. coli*. *S. cerevisiae* cultures and precultures have been performed based on a Verduyn mineral medium[49]. The precultures were performed in 1 L baffled flask overnight either at 37 °C (bacteria) or 30 °C (*S. cerevisiae*) at a shaking speed of 150 rpm and used to start the batch phase in a lab-scale stirred bioreactor (Biostat B-Twin, Sartorius) total volume: 2 L; working volume: 1 L at an initial $OD_{600}$ of 0.5. Once the batch phase was over (typically after 5, 8, and 15 h for *E. coli, B. subtilis,* and *S. cerevisiae,* respectively). The operating conditions used for the various cell systems are provided in Supplementary Table 2.

Data were collected through online FC during the experiments, and in Segregostat experiments, the actuator was pulsed based on both the observed distribution and a pre-defined set-point. Segregostat platform has been described earlier[10]. Briefly, every 12 min, a sample is automatically taken from the bioreactor, diluted, and analyzed in a flow cytometer (BD Accuri C6, BD Biosciences) with an FSC-H analysis threshold of 20,000 for bacteria and 80,000 for *S. cerevisiae*. During the chemostat experiment, glucose and, if applicable, an alternative carbon source was fed together, while in Segregostat experiment, glucose was the sole carbon source in the feed. In Segregostat, a feedback control loop, which includes a custom MATLAB script based on FC data, activates a pump to pulse an actuator. For *S. cerevisiae, E. coli* stress response, and *B. subtilis* sporulation, the actuator pulses glucose, while for *E. coli* alternative carbon source utilization, the actuator pulses lactose and arabinose. For all systems except for the sporulation control, a control threshold of 50/50 was utilized. However, for the sporulation control, the regulation was triggered once the fluorescence threshold was exceeded by more than 20% of the cells. This decision was made based on the irreversibility of the sporulation process, which required earlier intervention.

## Microfluidic cultivations and time lapse microscopy

For the experiments conducted in classical MSSC devices, settings and conditions were the same as previously reported[58]. Specifically, for the experiments conducted in MSCC device with constant environmental conditions, cells have been cultivated in microfluidic chips provided by Alexander Grünberger's lab (chambers size: 80 μm x 80 μm × 850 nm) in Verduyn medium with different glucose concentrations (5 μM, 0.1 mM, 0.2 mM, 0.4 mM, 0.6 mM, 1 mM and 3 mM, see Supplementary Fig. 2). In a second set of experiments, *S. cerevisiae* cells have been cultivated in the dynamic microfluidic single-cell cultivation (dMSCC) chips provided by Alexander Grünberger's lab (reference 24W, chambers size: 80 μm × 80 μm × 850 nm)[40]. Their design enables the simultaneous use of two cultivation media. They are separated into three zones: two control zones, fed by either one or the other medium, and a switching zone, fed in alternate by the two media. Diverse combinations of Verduyn medium with different glucose concentrations have been evaluated (i.e., 0.4–0.6 mM, 0.2–0.8 mM, 0.1–1 mM and 5 μM–3 mM). To approach Segregostat conditions, the duration of the switching zone feeding with the medium containing more (or less) glucose corresponds to the mean period with (or without) pulsing when population is controlled in Segregostat (i.e., 180 min (or 48 min)). High precision pressure pumps (line-up series, Fluigent, Le Kremlin-Bicêtre, France) were used to precisely control medium flow rate. The temperature was set at 30 °C. The chambers were inoculated with one or two cells by flushing the device with a cell suspension (OD600 between 0.4 and 0.5). At least six cultivation chambers were selected manually for each zone of the dMSCC chips. Microscopy images were acquired during 72 h using a Nikon Eclipse Ti2-E inverted automated epifluorescence microscope (Nikon Eclipse Ti2-E, Nikon France, France) equipped with a DS-Qi2 camera (Nikon camera DSQi2, Nikon France, France), a 100× oil objective (CFI P-Apo DM Lambda 100× Oil (Ph3), Nikon France, France). The GFP-3035D cube (excitation filter: 472/30 nm, dichroic mirror: 495 nm, emission filter: 520/35 nm, Nikon France, Nikon) was used to measure GFP. The phase contrast images were recorded with an exposure time of 300 milliseconds and an illuminator's intensity of 30%. The GFP images were recorded with an exposure time of 500 ms and an illuminator's intensity of 2% (SOLA SE II, Lumencor, USA). During the first 48 h, GFP and phase contrast images were acquired every hour. Phase contrast images were acquired every 6 minutes and GFP images every hour, for 24 h. The optical parameters and the time-lapse were managed with the NIS-Elements Imaging Software (Nikon NIS Elements AR software package, Nikon France, France). Single-cell data have been computed for the 24 last hours of the time lapse for at least 3 chambers per condition (i.e., zone of the dMSCC device). The cell-segmentation of the images and the measure of single-cell mean GFP intensity were performed using either the Python GUI[59] or the Matlab algorithm of Wood and Doncic[60]. This last was also used for cell tracking. In this case, seeds for segmentations have been corrected manually and at least ten cells per microfluidic cultivation chamber (well segmented and tracked all the time-lapse long) were selected manually.

## Determination of the impact of phenotypic switching on growth rate

To evaluate the difference in growth rate from the un-induced phenotype to the induced one, we have performed triplicate batch experiments in 1 mL-scale in a micro bioreactor (BioLector 2, m2p-labs, Baesweiler, Germany), using 48-well flower plates (MTP-48-B, Beckman Coulter Life Sciences, USA). The maximal growth rate was computed for *E. coli* W3110 and BL21 in the above described media with 5 g/L of glucose, *E. coli* W3110 P$_{araB}$::GFPmut2 in arabinose 5 g/L and *E.coli* W3110 P$_{lacZ}$::GFPmut2, *E. coli* BL21 pET28:GFP in lactose 5 g/L. The cultivations were performed with buffered media (MOPS 10 g/L) at 37 °C and with a shaking frequency of 1000 min$^{-1}$. All system experienced a growth reduction upon induction but only the induced phenotype of *E. coli* BL21 pET28:GFP had a growth rate below the dilution rate utilized in the continuous cultivation.

## Computation of the MI

MI is a proxy of information theory that describes the reduction in entropy conveyed by a specific external stimulus. We have computed MI for a system exhibiting a low switching cost i.e., the P$_{araB}$:GFP system in *E. coli* W3110, and for a system exhibiting a high switching cost i.e., the P$_{glc3}$:GFP system in *S. cerevisiae*. The computation of MI is based on the analysis of entropy at different defined environmental conditions, the response function. For the P$_{araB}$:GFP system, we have designed *E. coli* W3110 *ΔaraBAD* P$_{araB}$::GFPmut2, which is unable to consume arabinose, to characterize the relationship between the inducer concentration and induction profile. For knockout experiments, pTarget was modified with the Fw_sgRNA_20N_Ara primer and the homologous product was constructed from the upstream and downstream fragments generated with Fw_Frag1, Rv_Frag1, FW_Frag2, Rv_Frag2 (Supplementary Table 1)[56]. The deletion of the *araBAD* genes has been done to ensure a perfectly defined concentration of arabinose i.e., no consumption during the trial. From an overnight pre-culture, 10 flasks (100 mL total volume, 10 mL working volume) with buffered (10 g/L MOPS) mineral salt media were inoculated at an OD$_{600}$ of 0.5. Once the cells were in glucose limited conditions, a solution of arabinose was added to a final concentration ranging from 0 to 2 g/L. Eight concentrations in duplicate were analyzed. Following a delay of 24 min, samples were analyzed by FC (Supplementary Fig. 6) to determine the induction profile relative to the inducer concentration, the conditional entropy. Thus, the response function is the relation the links the population entropy to the stimulus concentration. As we could not prevent yeast from consuming glucose, the response function of the P$_{glc3}$:GFP in *S. cerevisiae* was determined by growing culture at different dilution rates in chemostat (Supplementary Fig. 7). The dilution rate of a chemostat sets the glucose uptake rate and was progressively increased to release the stress response of the population. This procedure is known as accelerostat (A-stat). In our case, the pump flow rate was modified with a step change every 2 h, resulting in a progressive increase of the dilution rate of 0.002 h$^{-1}$. This incremental range was chosen in order to ensure pseudo steady-state for each increment. The entire process was followed by automated FC for mapping the GFP distribution of the cell population (Supplementary Fig. 7).

From the response functions, MI was computed by subtracting the conditional entropies to the sum of all recorded entropies (this represents the space of all phenotypes that can be adopted by our system). An example of such computation is provided in Supplementary Note 3 and Supplementary Figs. 8–10.

## Modeling cell population dynamics based on FlowStocKS

The aim of this simulation toolbox is to be able to represent with high fidelity the population snapshots and dynamics captured based on automated FC. Briefly, population dynamics is modeled based on a set of ODEs representing the time evolution of biomass and substrates according to a Monod kinetics in a continuous cultivation device. From the global population, a given number of cells (approx. 10,000, some cells being washed-out during the simulation) are considered for generating a stochastic process. For these cells, phenotypic switching is modeled according to a Markov chain process driving the synthesis and degradation of GFP. Cell growth and division are considered for computing the GFP content. Upon switching, cells may meet a fitness cost depending on an inhibitory kinetics. The data are then fitted to a seven-decade fluorescence scale to fit to the automated FC data. Detailed information, including parameters and equations settings, are supplied in Supplementary Note 4 and Supplementary Table 3.

## Reporting summary

Further information on research design is available in the Nature Portfolio Reporting Summary linked to this article.

## Data availability

Data supporting the findings of this work are available within the paper and its Supplementary Information files. A reporting summary for this Article is available as a Supplementary Information file. Source data are provided with this paper.

## Code availability

The FlowStoCKS toolbox is available at GitLab.

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

## Acknowledgements

LH and VV are supported by an FRIA grant provided by the "Fonds de la Recherche Scientifique" FRS–FNRS from the Walloon region of Belgium. JAM is supported by a post-doctoral grant provided by the Service

Public de Wallonie (SPW) and the H2020 program of the European commission (Era-Cobiotech project Contibio). MD is supported by a PhD grant provided by the FRS–FNRS and the H2020 program of the European commission (Era-Net Aquatic Pollutant project ARENA). FD received funding from a research grant provided by the Service Public de Wallonie (SPW) and the H2020 program of the European commission (Era-Cobiotech project ComRaDes).

## Author contributions

L.H. performed the experiments with *E. coli* and *B. subtilis* and developed FlowStocKS. J.A.M. improved the FlowStoCKS codes and performed experiments related to *S. cerevisiae*. VV improved FlowStoCKS and the online FC data treatment procedures. M.D. performed the microfluidics experiments. S.T. and A.Z. prepared all the bioreactor experiments and designed the online FC interface. A.G. performed the microfluidics experiments and reviewed the paper. F.D. managed the researches, drafted the paper, and set the information theory computations.

## Competing interests

The authors declare no competing interests.
