## [Peer Review File · Nature Communications]

Fitness cost associated with cell phenotypic switching drives population diversification dynamics and controllabilityReviewers' Comments:

Reviewer #1:

Remarks to the Author:

This paper investigates the heterogeneity of the response of decision systems for cell populations grown either in constant or in time varying environments.

This paper is very original on two counts. First, it considers a very diverse set of decision systems, including several microbial species (*E. coli*, *B. subtilis*, *S. cerevisiae*) and both metabolic regulation and cell differentiation systems. Second, the authors define switching times for the environment based on the proportion of cells that have reached a given phenotype. Therefore, the switching frequency is adapted by construction to the typical switching time scale of the observed system. They call this mode of operation the *segregostat*, by analogy with the chemostat. This necessitates real-time flow cytometry capabilities and software/hardware integration.

The authors use standard notions from information theory (entropy, mutual information) to quantify the cell population heterogeneity. Based on experimental results, they propose a classification of the "diversification profiles" of their systems into three types: "constrained", "dispersed" and "bursty". They comment the impact of the cultivation mode of these three classes of systems on the population heterogeneity. Interestingly, the authors establish a connection between the type of the diversification profile and the fitness cost of switching.

On the positive side, one should stress that the research is very original, both regarding the question and the methodology used. The data is rich. The findings are interesting. Importantly, a model is able to recapitulate the main observations. The topic is connected to work and notions present in several communities (expression systems and bioproduction, cell decisions and fitness, information theory).

On the negative side, the text is sometimes hard to read and the presentation of the research material is not always optimal. Part of the challenge comes precisely from the fact that many notions are used from different fields. However, I feel that the clarity can be improved by a more gradual and rigorous introduction of the different problems addressed and notions used. Similarly, for the presentation of the results, a more precise description of the biological material and of the experimental setup used appears essential.

In summary, this paper contains highly original and interesting results. It can be of interest for researchers of several communities. Although in its present form, I do not consider it suitable for publication in *Nature Communications*, I think that it can be suitably revised to meet the standards of the journal.

Below I provide detailed comments and questions.

- L71: Please define the notion of population diversification regime
- L90: Environmental forcing triggered by cell switching dynamics leads to coordinated gene expression for diverse biological systems": this might appear a paradox to have the environment switches that depend on the cell switching dynamics. Clarify this point in a more precise manner.
- L95: "a methodology to compute the fluxes of cells from a phenotype to another": phenotypes are defined as discrete values. The definition of phenotype needs to be provided. At this stage, we only know that "GFP reporter bearing strains" are used.
- L101. "the cumulative probabilities of occurrence": cumulative?
- L149: "A feature of these two systems is that their activation does not result in a reduction in growth rate". This data is not shown. It would be interesting to show it in Figure 2 for example.

- L150: The switching cost, or fitness reduction, is defined as a reduction in growth rate. But how to define it properly when using a change in carbon source to switch the phenotype... The reductions in growth rate coming from the switching cost itself and coming from the change of carbon source will be convoluted together.

- L190 and following. The computation of the mutual information is not very clear. It seems that one needs to know the glucose uptake rate. The latter is "determined based on the value of the dilution rate, as well as based on glucose and biomass measurements." But no material and methods are provided for these computations.

- L246: "we applied environmental fluctuations between 0.1 and 1 mM of glucose, at the frequency recorded in the Segregostat conditions". Could you comment about the effective glucose profiles perceived by cells in segregostat. Are they comparable? Moreover, changes are abrupt in dMSCC and smooth and slow in bioreactors.

Figure 4B: how mean and variance compare with the data of figure 2D?

- L235. "However, when the nutrient level is changed at a given frequency, switching can be kept under control, leading to a drastic reduction of the phenotypic heterogeneity of the cell population." The heterogeneity of Fig 4B is compared to what data exactly to support this claim? We would need similar data taken from experiments shown in conditions 4A, no?

In Figure 5, diagrams at the top are not the correct ones (lactose, glucose).

In the P_T7:GFP system, what is pulsed? Same question for the spoIIE system? We have this data in table S3 only. It should be better integrated to the main text. Also, for the T7 system, we know that cells are "E. coli BL21 (DE3) carrying pET28:GFP (Ref56)". How to understand the connection with glucose pulses without reading the cited reference? More generally, the experimental systems should be described in (much) more details. and not only in the M&M section.

- L276: "Then, upon environmental forcing based on Segregostat cultivation, the number of bursts is reduced, and the fluxes of cells involved in the process are increased, leading to a substantial but temporary reduction of the entropy for the population.": OK but this is not very apparent from the plots, since they have different scales for the y-axis.

- L286: "In the context of the T7 system, this approach led to the maximization of cells in the GFP positive state suggesting that it could be used for mitigating metabolic burden and maximizing productivity in continuous bioprocesses.": This is not obvious from the data. Could you provide a more quantitative estimate?

- L328: "based FlowStockS" -> based on FlowStockS

Figure 7F: the reduction of entropy appears to be small comparatively to the entropy. Smaller than in the experimental data. This is not a problem, but a comment might be useful.

Equation 3: μ is an increasing function of substrate (which is expected) and an increasing function of the inhibitor (which is not expected). Maybe somewhat connected with equation 7 and the fact that n is negative in the parameter list...

Model presentation could be improved. For example, it seems that two representations of cells are used jointly (biomass and individual cells). This could be commented. Also why do you use "a population of 10,000 cells, each with an initial mass equal to 1/1000th of the initial biomass." Is it a typo?

Table S3 should be integrated in the main text so that one better understands the experiment that are performed.

Reviewer #2:

Remarks to the Author:

In this work, the authors explore the link between phenotypic diversification and fitness cost in yeast and bacteria. Using quantitative experiment and a numerical model, the authors propose that the way cells adapt to stress by switching from one phenotype to another can be clustered into three behaviors (constrained, diversified, bursty) with increasing fitness cost during the phenotypic switch. Classically, phenotypic diversification can be observed in chemostat or batch when a resource becomes limiting. Here, the authors also developed a device to automatize the generation of diversification cycles by using a feedback loop control method, called Segregostat. It switches the external stress linked to the phenotypic diversification in function of the population phenotypic structure. The population structure is monitored by FACS automated measurements and the author use basics of information theory (entropy and mutual information) to follow the diversity of the population. Taken together, the article is interesting and can reach a large audience of biologists (evolution, synthetic biology, bioproduction) but, in its present form the article is hard to grasp and should be significantly improved before it can be considered for publication.

Without begin exhaustive, here are a few suggestions to improve the readability of this article. For instance, the definition of entropy and mutual information (and how it is used) should be described in the main text (and not in the supplementary information). The authors should also better explain why they are interested in mutual information (entropy is probably enough to distinguish between the three diversification scenario). Indeed, Figure 3 is very hard to understand. May be the authors should explain why it is important to compare the mutual information from the four different systems? The Figures may be clarified and be explicit on which system and which environment are used / switched and at which level. Time axis should be in "hours" not in "time interval #". Importantly, how reproducible such experiments and measurements of entropy and mutual information are? Can the author compare several replica experiments using the segregostat? This could be shown in supplementary materials. The supplementary movies do not bring much information and I do not think they are needed. The author mention the notion of controllability of the system, but of which controllability are they making reference to? The controllability of the phenotypic switching? Of the global population structure? Of the gene of interest? The discussion on the single cell fitness measurements is a bit out of place and could be moved to supplementary information to keep a focus on segregostat / chemostat comparison.

Reviewer #3:

Remarks to the Author:

The authors address an interesting research topic in this paper, examining how the diversification dynamics of gene expression pathways in bacteria and yeast and the fitness cost of switching into a different state are coupled. They combine single cell approaches with stochastic modelling to reveal three classes of diversification regimes, from constrained at low switching cost to bursty for high switching cost. In general, the amount of data in the paper in the paper is impressive and the analysis is interesting. I find the paper to be of general interest but some of the presentation could be improved to enhance understanding. I have the following major and minor points

Major Points

The evidence that the classes of diversification are linked to fitness cost of switching could be further strengthened. This is a major claim of the paper, but it is a bit unclear if there are any other possible classes that could exist and if the classes are only due to the fitness cost, and not due to other

differences between the experimental systems. Is there some way to firm up a little further the link to fitness cost in the experimental data, given that this is one of the major claims of the paper? Could you for example reduce the fitness cost of switching into a Pglc:GFP ON state by adding in additional mutations to the system, or using different environmental conditions where the fitness cost is less, and see if it follows the model predictions? Alternatively perhaps the fitness cost of the T7 system could be more easily tuned? This would be a more direct test of the model than looking at multiple systems as you would be able to transition between the different states in the same system?. Also, the biological significance of the three classes of diversification schemes could be discussed in a little more detail in the discussion.

Reproducibility: There is a lack of information of number of repeats or number of cells in the figures. Can the authors spell out the number of cells as well as the biological replicates carried out?

Minor

Figure 1 – how are the 45 bins chosen for the fluorescence? There are 50 bins in figure 2? It would be helpful to have y axis values on this figure and figure 2 for the bins?

Figure 2- it is a little confusing how the on and off states in the cartoon match the pulse states in the diagram. For C and D, Are you pulsing in Glucose or pulsing out? I am confused because it looks like you are pulsing in Glucose, but the bolA expression appears to go up with the increase in glucose rather than down? Could you label the Pulse diagram on each part of the figure to explain what the levels are going from and to? Similarly it is hard to see what is going on in the expression of in Pglc3:GFP in Fig 2D. there seems to be blank vertical lines in the data, and it is a little difficult to interpret the distributions. Finally, is there something interesting that could be said about the pulsing frequency for the different systems, in terms of the diversification dynamics?

Figure 4 B – the switching frequencies between 1mM and 0.1mM Glucose are meant to match that of Figure 2D, but I am struggling to see how they match? Also the expression peaks after the removal of the 0.1mM Glucose, is this due to the maturation time of the GFP?

Sup Figure 1 – the y axis label is covered up for some of the panels. Also, maybe spell out again what the blue/red mean or refer to where first defined?

Typo, l 463, previsouly

Point-by-point responses for the proposal “Fitness cost associated with cell phenotypic switching drives population diversification dynamics and controllability”

Dear reviewers,

First of all, we would like to thank you all about the very constructive comments you have made and the time taken for reviewing our proposal. As requested, we have prepared below a point-by-point response to all the comments that have been raised. We have also highlighted in yellow all the changes made at the level of the manuscript.

Reviewer 1

- L71: Please define the notion of population diversification regime

The population diversification regime depends on the mode of phenotypic switching i.e., either responsive or stochastic, used by individual cells within the population. We have now reorganized the paper in order to make this notion more appealing for the reader. A specific item has been added in the introduction section (L60) and the first result section has been modified in order to explain the methodology behind the determination of the population diversification regime, as well as the corresponding quantitative features such as the entropy profile $H(t)$ (new section, L78, “Characterization of population diversification dynamics based on automated flow cytometry”). Also, Figure 1 has been adapted for better illustrating the notion of “population diversification dynamics”. Finally, we end up with the definition of the different cell population diversification regimes i.e., either constrained, dispersed or bursty, that is given in the next results sections.

- L90: Environmental forcing triggered by cell switching dynamics leads to coordinated gene expression for diverse biological systems": this might appear a paradox to have the environment switches that depend on the cell switching dynamics. Clarify this point in a more precise manner.

This result is indeed not intuitive. We have now changed the title in order to make clear that a cell-machine interface is needed for applying environmental fluctuations matching cell switching dynamics.

- L95: "a methodology to compute the fluxes of cells from a phenotype to another": phenotypes are defined as discrete values. The definition of phenotype needs to be provided.

At this stage, we only know that "GFP reporter bearing strains" are used.

The phenotype is now described at L85 "We define the phenotype as the cellular content in GFP and the characteristics associated with the activation of the observed gene circuit".

- L101. "the cumulative probabilities of occurrence": cumulative?

This term has been removed and the binning procedure, as well as its result in term of entropy and cellular fluxes, are explained in the first section of the Results (new section).

- L149: "A feature of these two systems is that their activation does not result in a reduction in growth rate". This data is not shown. It would be interesting to show it in Figure 2 for example.

We have now performed new experiments in order to determine the growth rate of cells upon phenotypic switching. All the data are shown in Figure S5.

- L150: The switching cost, or fitness reduction, is defined as a reduction in growth rate. But how to define it properly when using a change in carbon source to switch the phenotype... The reductions in growth rate coming from the switching cost itself and coming from the change of carbon source will be convoluted together.

The switch from glucose to arabinose or lactose for the strain E. coli W3110 doesn't lead to a significant reduction in growth rate, by comparison to the dilution rate that has been used during the cultivations. All these data are shown in Figure S5. But the reviewer has right, the effect of gene circuit activation and change of C source are coupled most of the time. However, we can see that in most of the cases, the activation of the gene circuit itself drives the switching cost. The data presented at Figure S5 illustrate quite well this effect. It can be seen that the E. coli W3110 strain exhibit slightly reduced growth when cultivated on lactose and arabinose, independently of the presence of a fluorescent reporter. On the opposite, we can see that the strain E. coli BL21(DE3) exhibit a huge reduction in growth upon cultivation on lactose, mostly due here to the activation of the chromosomal insert of the T7 RNA polymerase. These data exemplified the two cases i.e., a system exhibiting a low switching cost (E. coli W3110) and another one exhibiting a high switching cost (E. coli BL21(DE3)).

- L190 and following. The computation of the mutual information is not very clear. It seems that one needs to know the glucose uptake rate. The latter is "determined based on the value of the dilution rate, as well as based on glucose and biomass measurements." But no material and methods are provided for these computations.

As also reported by the other reviewers, the computation of mutual information is not clearly described in the proposal. We have now reorganized the paper in order to introduce more sequentially the different biological systems, as well as the different types of computation made based on population data. The concept of MI is now introduced later in the text (in the new section "Coordinated gene expression does not necessarily lead to a more homogeneous cell population") and is linked with the controllability of cell population. A specific section has also been added in the Material and Methods section.

- L246: "we applied environmental fluctuations between 0.1 and 1 mM of glucose, at the frequency recorded in the Segregostat conditions". Could you comment about the effective glucose profiles perceived by cells in segregostat. Are they comparable? Moreover, changes are abrupt in dMSCC and smooth and slow in bioreactors.

There are indeed differences between the glucose fluctuations generated in Segregostat and in microfluidics. These changes are abrupt and well controlled in microfluidics, whereas it is less controlled in the case of the Segregostat. However, since we are working with pulses of glucose, these changes are also pretty abrupt in the case of the Segregostat, allowing generating similar biological responses.

- Figure 4B: how mean and variance compare with the data of figure 2D?

We have now computed the entropy H for the microfluidics and the Segregostat data in the case of the Pglc3:GFP system. The entropy values are provided, both for MSCC and dMSCC experiments, at Figure 4F. We can see that switching periodically between 0.1 and 1 mM of glucose tends to uniformize the response of each cell within the population, in a way that is very similar to the one reported during the Segregostat experiments.

- L235. "However, when the nutrient level is changed at a given frequency, switching can be kept under control, leading to a drastic reduction of the phenotypic heterogeneity of the cell population." The heterogeneity of Fig 4B is compared to what data exactly to support this claim? We would need similar data taken from experiments shown in conditions 4A, no?

We agree with the reviewer. However, due to several comments made by the reviewers about the microfluidics data, the dMSCC data have now been integrated in a global figure (Figure 4) as a panel. The classical MSCC data have been moved to SI. To reply to the comment made by the reviewer, a panel at Figure 4E provide now a comparative analysis about the different microfluidics set-up. For the dMSCC data, we can observe that the standard deviation (shaded area) is lower than the one obtained when cultivating the cells under constant conditions with 0.1 mM of glucose.

- In Figure 5, diagrams at the top are not the correct ones (lactose, glucose). In the P_T7:GFP system, what is pulsed? Same question for the spoIIIE system? We have this data in table S3 only. It should be better integrated to the main text. Also, for the T7 system, we know that cells are "E. coli BL21 (DE3) carrying pET28:GFP (Ref56)". How to understand the connection with glucose pulses without reading the cited reference? More generally, the experimental systems should be described in (much) more details. and not only in the M&M section.

Figure 5 is now Figure 3 in the revised version of the paper. We apologize for the lack of clarity, but the pulsing strategy initially shown is correct. The pET28:GFP system is a T7-based expression vector inducible with lactose. When lactose is pulsed, cells are turning green. When lactose pulsing is stopped, GFP is diluted by growth (glucose is added continuously according to a classical continuous mode of cultivation). The PspoIIIE:GFP system is induced upon glucose limitation. In our case, this system is cultivated at a very low dilution rate (0.1

h⁻¹) in order to generate glucose limitation and stimulated cell switching to sporulation. When too much cells are switching, additional glucose is pulsed in order to keep population under control. We have now added these explanations to the text.

- L276: "Then, upon environmental forcing based on Segregostat cultivation, the number of bursts is reduced, and the fluxes of cells involved in the process are increased, leading to a substantial but temporary reduction of the entropy for the population.": OK but this is not very apparent from the plots, since they have different scales for the y-axis.

We have reorganized the figure introducing the T7 and spoII E systems exhibiting bursty diversification profile (now Figure 3 in the revised version). The scales are now comparable to each other.

- L286: "In the context of the T7 system, this approach led to the maximization of cells in the GFP positive state suggesting that it could be used for mitigating metabolic burden and maximizing productivity in continuous bioprocesses.": This is not obvious from the data. Could you provide a more quantitative estimate?

The data are now displayed in figure 3. The data are discussed in function of the flux of cells $F(t)$. We have also added a colorbar displaying the amount of cells for each bin/phenotypic state.

- L328: "based FlowStockS" -> based on FlowStockS
Correction has been made in the text

- Figure 7F: the reduction of entropy appears to be small comparatively to the entropy. Smaller than in the experimental data. This is not a problem, but a comment might be useful.

The data are now displayed in Figure 6F. The reduction in entropy is associated with the controllability of the system (controllability is defined as the gain in entropy when cell population is cultivated in Segregostat condition, by comparison with chemostat). The connection with controllability is done based on a statement in the legend of Figure 6 referring to the controllability results shown in Figure 4.

- Equation 3: μ is an increasing function of substrate (which is expected) and an increasing function of the inhibitor (which is not expected). Maybe somewhat connected with equation 7 and the fact that n is negative in the parameter list...

This is a typo from our part, μ is inversely proportionate to the inhibitor concentration. This has been corrected in the text, the correct equation was used in the model.

- Model presentation could be improved. For example, it seems that two representations of cells are used jointly (biomass and individual cells). This could be commented. Also why do you use "a population of 10,000 cells, each with an initial mass equal to 1/1000th of the initial biomass." Is it a typo?

It was indeed a typo. What is called “cells” in FlowStockS do represent a fraction of the biomass as the sum of all their individual biomass equals the total biomass of the system. We have corrected the typo and rephrased the description of the variables to avoid any confusion.

- Table S3 should be integrated in the main text so that one better understands the experiment that are performed.

We have decided not to include this table in the main text since we are already out of space. Instead, we have reorganized the text in order to introduce the 6 different biological systems in a more amenable way for the reader.

Reviewer 2

- The definition of entropy and mutual information (and how it is used) should be described in the main text (and not in the supplementary information). The authors should also better explain why they are interested in mutual information (entropy is probably enough to distinguish between the three diversification scenario). Indeed, Figure 3 is very hard to understand. May be the authors should explain why it is important to compare the mutual information from the four different systems? The Figures may be clarified and be explicit on which system and which environment are used / switched and at which level.

As also reported by the other reviewers, the computation of mutual information is not clearly described in the proposal. We have now reorganized the paper in order to

introduce more sequentially the different biological systems, as well as the different types of computation made based on population data. The concept of MI is now introduced later in the text (in the new section “Coordinated gene expression does not necessarily lead to a more homogeneous cell population”) and is linked with the controllability of cell population. A specific section has also been added in the Material and Methods section.

- Time axis should be in “hours” not in “time interval #”.

All the figures have been adapted according to this comment.

- Importantly, how reproducible such experiments and measurements of entropy and mutual information are? Can the author compare several replica experiments using the segregostat? This could be shown in supplementary materials.

The reproducibility of all the data has been assessed and is now shown in Figure S3.

- The supplementary movies do not bring much information and I do not think they are needed.

The movies have been moved to Supplementary information where a Gitlab link is provided to access them.

- The author mention the notion of controllability of the system, but of which controllability are they making reference to? The controllability of the phenotypic switching? Of the global population structure? Of the gene of interest?

We define the controllability as the difference in entropy between the chemostat and the Segregostat. Controllability is now clearly defined in the section “Coordinated gene expression does not necessarily lead to a more homogeneous cell population” and in Figure 4.

- The discussion on the single cell fitness measurements is a bit out of place and could be moved to supplementary information to keep a focus on segregostat / chemostat comparison.

We have now reorganized the results section in order to introduce sequentially the data about chemostat and Segregostat for the 6 different biological systems considered. The figure with the microfluidic data has been partially moved to SI and to the panels D and E of the new Figure 4.

Reviewer 3

- The evidence that the classes of diversification are linked to fitness cost of switching could be further strengthened. This is a major claim of the paper, but it is a bit unclear if there are any other possible classes that could exist and if the classes are only due to the fitness cost, and not due to other differences between the experimental systems. Is there some way to firm up a little further the link to fitness cost in the experimental data, given that this is one of the major claims of the paper? Could you for example reduce the fitness cost of switching into a Pglc:GFP ON state by adding in additional mutations to the system, or using different environmental conditions where the fitness cost is less, and see if it follows the model predictions? Alternatively perhaps the fitness cost of the T7 system could be more easily tuned? This would be a more direct test of the model than looking at multiple systems as you would be able to transition between the different states in the same system?

We thank the reviewer for this very relevant comment. Indeed, we have identified the fitness cost as a major driver of diversification but, as we have now acknowledged in the discussion section, other factors, such as the switching rate or the reversibility of

the switch, could also have an impact. Regarding the link between fitness cost and diversification regime, we have strengthened our claim by analyzing the switching cost of some of our case studies where it had not previously been done (Figure S5). Adding an additional case study has been considered, but we did not manage to find a system where we could easily tune the fitness cost. Working based on the T7 system is a good idea since we have now different split_T7 systems available. However, this system provided only an all-or-non response and it is not possible to progressively increase or decrease the expression of the T7 RNA polymerase. This is also the reason why we designed FlowStoCKS, a robust simulation tools, allowing to reproduce the time dependent FC profiles observed experimentally. Based on this model, we have been able to show that the switching cost is the main drivers for the different diversification regimes observed experimentally.

- Also, the biological significance of the three classes of diversification schemes could be discussed in a little more detail in the discussion.

We have now discussed the biological consequences of the diversification regime in the Discussion section. We observe indeed that the systems exhibiting a high switching cost display more stochasticity in their diversification regimes, as recorded by the higher entropy. There is approximately an additional entropy of 1 bit, by comparison with the systems exhibiting low or no switching cost. This extra entropy gives probably more resilience to the cell population cultivated in continuous mode which otherwise would lead to the wash-out of the system (as observed based on the FlowStockS simulations.

- Reproducibility: There is a lack of information of number of repeats or number of cells in the figures. Can the authors spell out the number of cells as well as the biological replicates carried out?

The number of biological replicates is now clearly indicated in the legend, as well as the number of cells analyzed per time interval. Reproducibility of the data has been assessed and the data are available in the supplementary figure S3.

- Figure 1 – how are the 45 bins chosen for the fluorescence? There are 50 bins in figure 2? It would be helpful to have y axis values on this figure and figure 2 for the bins?

It was indeed a mistake since 50 bins have been used for all the computations. We have now revised to whole paper to make this point consistent. It is not possible to provide y-axis value since we are considering bin numbers. Inside each bin, we have a specific number of cells. The relative number of cells between each bin for a time interval can be visualized by the colorbar.

- Figure 2- it is a little confusing how the on and off states in the cartoon match the pulse states in the diagram. For C and D, Are you pulsing in Glucose or pulsing out? I am confused because it looks like you are pulsing in Glucose, but the bolA expression

appears to go up with the increase in glucose rather than down? Could you label the Pulse diagram on each part of the figure to explain what the levels are going from and to? Similarly it is hard to see what is going on in the expression of in Pglc3:GFP in Fig 2D. there seems to be blank vertical lines in the data, and it is a little difficult to interpret the distributions. Finally, is there something interesting that could be said about the pulsing frequency for the different systems, in terms of the diversification dynamics?

Figure 2 has now been totally redesigned in order to avoid any misinterpretation. The pulsing bar has been now labelled in order to show the fluctuations in amplitude. Data for the Pglc3:GFP system have been zoomed (limited on the time axis). The pulsing frequency is different for each of the systems investigated, mainly due to the amount of GFP accumulated that are different, and then takes longer to be diluted. This observation has not been included in the text in order to focus of the cell-to-cell coordination in gene expression effect.

- Figure 4 B – the switching frequencies between 1mM and 0.1mM Glucose are meant to match that of Figure 2D, but I am struggling to see how they match? Also the expression peaks after the removal of the 0.1mM Glucose, is this due to the maturation time of the GFP?

This figure has now been deleted and the microfluidics data have been incorporated in the new Figure 4 as panel D. We used the same period as the one used in the Segregostat experiment i.e., 0.8 h in glucose limitation (0.1 mM) and 3h in glucose excess (1 mM).

- Sup Figure 1 – the y axis label is covered up for some of the panels. Also, maybe spell out again what the blue/red mean or refer to where first defined?

This figure is now Figure S2 and has been totally redesigned.

- Typo, l 463, previsouly

The typo has been corrected

Sincerely yours,

Reviewers' Comments:

Reviewer #1:

Remarks to the Author:

The structure of the paper has been significantly reorganized, with additional or improved figures, and the text has been expanded at places to better motivate and explain the notions that are employed. As a result, the article now has a logical flow that is much easier to follow. This represents a very significant improvement.

However, the article is still not very easy to read. But this may stem from the original angle of the research question addressed and the interdisciplinary nature of the work.

The answers to my remarks have been addressed satisfactorily.

I have a few very minor comments to make:

- lines 201 and 205: there are some repetitions
- in material and methods, the computation of the mutual information is still a bit hard to understand. In the paragraph, one describes how to characterize the relationship between the inducer concentration and induction profile (with no initial motivation) and concludes by "subtracting the conditional entropies from the total entropy" with no connection to what precedes... Introducing that mutual information (MI) is based on the response function of a cell population at the beginning of the paragraph, and connecting the two notions at the end of the paragraph should provide some clarity. This is nicely done in the supplementary material for example.
- in the supplementary material, page 6: y is not defined in Equation 2 (sugar uptake rate probably). Figure S8A: is the conditional entropy not a 2D function?

Reviewer #2:

Remarks to the Author:

Dear Authors, Dear Editor,

The revised version of the article improves significantly the readability and better convey the scientific interest of the segregostat and how it is used to establish the main findings of the article. I recommend publication of this article.

As a side note, several figures could still be improved to homogenize the size of the fonts in the figures and to improve readability. Also Figure 1 still mention "time interval #" which should be replaced by a time interval in hours.

Reviewer #3:

Remarks to the Author:

The authors have addressed my concerns.

Point-by-point responses for the proposal “Fitness cost associated with cell phenotypic switching drives population diversification dynamics and controllability”

Dear reviewers,

We would like to thank you for accepting this paper and for the effort you dedicated to thoroughly review our work. These comments have helped increase the quality of our message and in its spirit please find below a point-to-point response to your comments.

Reviewer 1

- lines 201 and 205: there are some repetitions

After having read the section again, we share your observation that some repetitions occur. The text has been modified to remove the repetition concerning the induction of the T7 expression system.

- in material and methods, the computation of the mutual information is still a bit hard to understand. In the paragraph, one describes how to characterize the relationship between the inducer concentration and induction profile (with no initial motivation) and concludes by "subtracting the conditional entropies from the total entropy" with no connection to what precedes... Introducing that mutual information (MI) is based on the response function of a cell population at the beginning of the paragraph, and connecting the two notions at the end of the paragraph should provide some clarity. This is nicely done in the supplementary material for example.

The description of the computation of mutual information in the material and method has been revised to more progressively introduce the notion mutual information, better connect it to the response function and explain the rationale behind the two set-ups to define these functions.

- in the supplementary material, page 6: y is not defined in Equation 2 (sugar uptake rate probably). Figure S8A: is the conditional entropy not a 2D function?

We have now defined the term in equation 2, which is indeed the sugar uptake rate. In Figure S8A, for each sugar uptake rate value, each corresponding GFP distribution have been processed in order to compute the conditional entropy, leading to a 1D function.

Reviewer 2

Several figures could still be improved to homogenize the size of the fonts in the figures and to improve readability. Also Figure 1 still mention "time interval #" which should be replaced by a time interval in hours

A new version of the figure 1 is proposed with improved graphics and where the label "time interval #" has been changed for time interval in hours. Also, some fonts in the figure 2, 3, 4 and 6 have been adapted to improve the readability.

Sincerely yours,

Frank Delvigne